# Central Asian Rodents as Model Animals for *Leishmania*
*major* and *Leishmania donovani* Research

**DOI:** 10.3390/microorganisms8091440

**Published:** 2020-09-20

**Authors:** Barbora Vojtkova, Tatiana Spitzova, Jan Votypka, Tereza Lestinova, Iveta Kominkova, Michaela Hajkova, David Santos-Mateus, Michael A. Miles, Petr Volf, Jovana Sadlova

**Affiliations:** 1Department of Parasitology, Faculty of Science, Charles University, 12844 Prague, Czech Republic; barbora.vojtkova@natur.cuni.cz (B.V.); tatiana.spitzova@gmail.com (T.S.); vapid@natur.cuni.cz (J.V.); terka.kratochvilova@seznam.cz (T.L.); ivetkominek@gmail.com (I.K.); volf@cesnet.cz (P.V.); 2Department of Cell Biology, Faculty of Science, Charles University, 12844 Prague, Czech Republic; michaela.hajkova1@gmail.com; 3Department of Infection Biology, Faculty of Infectious and Tropical Diseases, London School of Hygiene and Tropical Medicine, London WC1E 7HT, UK; David.Mateus@lshtm.ac.uk (D.S.-M.); Michael.Miles@lshtm.ac.uk (M.A.M.)

**Keywords:** *Phodopus sungorus*, *Cricetulus griseus*, *Lagurus lagurus*, model animals, cutaneous leishmaniasis, visceral leishmaniasis, infectiousness, xenodiagnosis, *Phlebotomus*, sand fly

## Abstract

The clinical manifestation of leishmaniases depends on parasite species, host genetic background, and immune response. Manifestations of human leishmaniases are highly variable, ranging from self-healing skin lesions to fatal visceral disease. The scope of standard model hosts is insufficient to mimic well the wide disease spectrum, which compels the introduction of new model animals for leishmaniasis research. In this article, we study the susceptibility of three Asian rodent species (*Cricetulus griseus, Lagurus lagurus,* and *Phodopus sungorus*) to *Leishmania major* and *L. donovani.* The external manifestation of the disease, distribution, as well as load of parasites and infectiousness to natural sand fly vectors, were compared with standard models, BALB/c mice and *Mesocricetus auratus*. No significant differences were found in disease outcomes in animals inoculated with sand fly- or culture-derived parasites. All Asian rodent species were highly susceptible to *L. major*. *Phodopus sungorus* showed the non-healing phenotype with the progressive growth of ulcerative lesions and massive parasite loads. *Lagurus lagurus* and *C. griseus* represented the healing phenotype, the latter with high infectiousness to vectors, mimicking best the character of natural reservoir hosts. Both, *L. lagurus* and *C. griseus* were also highly susceptible to *L*. *donovani,* having wider parasite distribution and higher parasite loads and infectiousness than standard model animals.

## 1. Introduction

*Leishmania* parasites (Kinetoplastida: Trypanosomatidae) are causative agents of leishmaniases, a group of diseases prevalent worldwide in 98 countries with more than 350 million people considered at risk, more than 1 million new cases occurring every year, and more than 50 thousand deaths annually, due to the visceral form [1,2]. The epidemiology and ecology of leishmaniases are exceptionally complex—at least 20 *Leishmania* species are pathogenic to humans, each of them possessing different mammalian reservoir hosts, as well as insect vectors [3].

This species diversity is reflected in the broad spectrum of clinical manifestations of human leishmaniases, which results from interactions between the parasite species and the host immune responses. Cutaneous leishmaniasis (CL), transmitted by diverse sand fly vector species, is caused principally by *L. major*, *L. tropica*, and *L. aethiopica* in the Old World and *L. mexicana*, *L. venezuelensis*, *L. amazonensis*, *L. braziliensis*, *L. panamensis*, *L. guyanensis,* and *L. peruviana* in the New World. Infection with CL may be characterized by localized, diffuse, or disseminated skin lesions [4]. Metastatic mucocutaneous leishmaniasis (MCL), confined to the New World, is due to *Leishmania (Viannia) braziliensis* or, less frequently, *L. (V.) panamensis* and *L. (V.) guyanensis.* Visceral leishmaniasis (VL), caused by *L. donovani* and *L. infantum*, is the most severe form, often fatal if left untreated, characterized by fever, loss of weight, splenomegaly, hepatomegaly and/or lymphadenopathy, and anemia [5].

Experimental animal models are expected to mimic the specific features of the variety of human leishmaniases. Many immunological aspects of the disease have been studied using standard laboratory models, such as mice, hamsters, domestic dogs, and non-human primates. However, none of them accurately reproduces the outcome of human *Leishmania* infection [6]. The major advantage of inbred mouse models is their controlled genetic background and well-defined immune response. On the other hand, replication and spread of the pathogen in mice are far from the natural pattern. The relative lack of genetic polymorphism in laboratory mice has been specifically overcome by using stocks derived from recently trapped wild progenitors belonging to different taxa of the genus *Mus* [7]. An alternative approach is the use of genetically polymorphic wild rodents as experimental animal models for host-parasite relationships studies. These models allow a better understanding of the dynamics and range of infection, including mechanisms of parasite amplification, their availability for transmission, and the natural regulation of the immune response [6].

More than 20 rodent species have been used for experimental infections with *Leishmania* parasites. The extensive research in this field was mainly done during the first half of the last century. However, many of the tested species are protected or difficult to breed in the laboratory, and many of them have been shown to be resistant to infection (reviewed in [8]). More recently, two New World wild rodent species *Thrichomys laurentius* and *Peromyscus yucatanicus* have been used for the study of *L. infantum*, *L. braziliensis*, and *L. mexicana* infections, respectively [9,10].

Our study was aimed at evaluation of susceptibility of three Central Asian rodent species, *Cricetulus griseus*, *Phodopus sungorus,* and *Lagurus lagurus* to two human pathogenic *Leishmania* species. The three rodent species were chosen as they are commercially available and relatively easy to breed. Moreover, several publications from the beginning of the last century showed the susceptibility of *C. griseus* to *Leishmania donovani* [11,12,13,14], while *L. lagurus* was used for testing of VL in China (reviewed in [15]). Here, rodents were infected with *L. major* and *L. donovani,* using different inoculum types (sand fly vs. culture-derived parasites). Fluorescence detection and quantitative PCR were applied to evaluate the load of parasites and their anatomical distribution in the rodent.

## 2. Materials and Methods 

### 2.1. Sand Flies, Parasites, and Rodents

The sand fly colonies of *Phlebotomus duboscqi* (originating from Senegal) and *Phlebotomus orientalis* (originating from Ethiopia) were maintained in the insectary of the Department of Parasitology, Charles University in Prague, under standardized conditions (26 °C, fed on 50% sucrose, and with 14 h light/10 h dark photoperiod) as previously described [16].

Two *Leishmania* strains were used for infection of sand flies and rodents: *L. major* (MHOM/IR/-/173) and *L. donovani* (MHOM/ET/2009/AM459) with DsRed and RFP red fluorescent markers, respectively, prepared according to the methodology described by Sadlova et al. [17]. Promastigotes were cultured in M199 medium (Sigma-Aldrich, St. Louis, MO, USA) containing 10% heat-inactivated fetal calf serum (FBS Gibco, Thermo Fisher Scientific, Waltham, MA, USA), 2% sterile human urine, supplemented with 1% BME vitamins (Basal Medium Eagle, Sigma-Aldrich, St. Louis, MO, USA) and 250 µL/mL amikacin (Medopharm, Prague, Czech Republic).

Five rodent species were used: *Cricetulus griseus* (Chinese hamsters), *Lagurus lagurus* (Steppe lemmings), *Phodopus sungorus* (Djungarian hamsters)*, Mesocricetus auratus* (Golden hamsters), and *Mus musculus* (BALB/c mice), all originating from commercial sources. BALB/c mice and *M. auratus* originated from AnLab s.r.o. (Prague, Czech Republic), *C. griseus, Phodopus sungorus*, and *L. lagurus* from Karel Kapral s.r.o. (Prague, Czech Republic). Breeding containers T2 and T3 (Velaz s.r.o., Prague, Czech Republic) were used for maintaining BALB/c mice; the remaining species were maintained in breeding containers 3/H3 and T4 (Velaz s.r.o., Prague, Czech Republic). Containers were equipped with German Horse Span bedding (Pferde a.s., Prague, Czech Republic), breeding material (Woodwool, Miloslav Vlk s.r.o., Ratibor, Czech Republic), hay (Krmne smesi Kvidera s.r.o, Nezvestice, Czech Republic), and the standard feed mixture ST-1 (Velaz s.r.o., Prague, Czech Republic). The feed mixture MOK (Biopharm s.r.o., Kraluv Dvur, Czech Republic) without animal protein content was used for *L. lagurus*. Food and water were provided *ad libitum*, and the animals were maintained in standard conditions (12 h dark/12 h light photoperiod, at a temperature of 22–25 °C and humidity of 40–60%).

### 2.2. Experimental Infections of Sand Flies

Promastigotes from log-phase cultures were resuspended in heat-inactivated defibrinated rabbit blood at a concentration of 1 × 10^6^ promastigotes/mL. Sand fly females were infected by feeding through a chick-skin membrane, and engorged specimens were maintained in the same conditions as the colony.

### 2.3. Infections of Rodents

Two methods for infection of rodents were used: Infections initiated with sand fly-derived promastigotes (SDP) according to Sadlova et al. [18] and infection initiated with stationary phase culture-derived promastigotes (CDP), to include infective metacyclic forms. For the first method, sand fly females experimentally infected with *Leishmania* parasites were dissected on day 8 post bloodmeal, when mature infections and accumulation of metacyclic forms had developed in the thoracic midguts (TMG). Pools of 80 freshly dissected TMG with a good density of parasites were homogenized in 40 µL of sterile saline. For the inoculation of rodents with CDP, stationary-phase culture was washed twice in saline. For the selection of *L. major* metacyclic forms, 50 µL/mL of peanut agglutinin lectin (PNA) was added, which agglutinated the procyclic forms with galactosyl side-chains of LPG. The non-agglutinated metacyclic forms were then separated using centrifugation (200 G/5 min, 4 °C) [19]. Parasites were counted using a Bürker chamber, and pools of 10^6^ or 10^8^ promastigotes (*L. donovani*), or 10^6^ metacyclic forms (*L. major*) were resuspended in 50 µL of saline. Dissected salivary glands of *P. duboscqi* and *P. orientalis* females (SG) were pooled (10 glands per 10 µL of saline) and stored at −20 °C. Prior to the inoculation of mice, SG were frozen in liquid nitrogen and thawed three times, and the resulting lysate was added to the inoculum.

Rodents were anesthetized with a mixture of 66 mg/kg ketamine and 26 mg/kg xylazine. Then, 5.5 µL of inoculum (mixed parasites with SG suspension) were injected intradermally into the ear pinnae. Therefore, the inoculum dose per animal with CDP comprised of 10^5^ or 10^7^ promastigotes (*L. donovani*) or 10^5^ selected metacyclic forms (*L. major*). Numbers of SDP were calculated using a Bürker chamber, and the proportion of metacyclic forms was identified on Giemsa stained smears based on morphological criteria described previously [20]. *Leishmania major* numbers ranged from 5.6 × 10^4^ to 7.6 × 10^4^ per animal with metacyclics comprising 69–73% of all forms, *L. donovani* numbers ranged from 5.0 × 10^4^ to 6.39 × 10^5^ per animal with 29–46% of metacyclic forms. Post inoculation (p.i.), rodents were checked weekly for external signs of the disease and killed at week 9–15 p.i. or 30 p.i. for groups infected with *L. major* and *L. donovani,* respectively. The experiments were performed in two runs in all rodent species.

### 2.4. Xenodiagnosis

Five to seven-day-old sand fly females were used for xenodiagnosis experiments. *Phlebotomus duboscqi* females (natural vectors of *L. major* [21]) were allowed to feed on the site of inoculation of *L. major* (ear pinnae) of anesthetized rodents; the rodent bodies were covered with a cotton bag, so that only the left ear pinnae were accessible to sand flies. *Phlebotomus orientalis* females (natural vectors of *L. donovani* [22]) were allowed to feed on the whole body of the anesthetized rodents infected with *L. donovani (*the smaller ear pinnae of *L. lagurus* did not produce sufficient access to initiate the feeding behavior of *P. orientalis* females). The rodents were placed into a small cage (20 × 20 × 20 cm) and 30–40 sand fly females were allowed to feed for one hour. The fed sand fly females were separated and maintained at 26 °C on 50% sucrose. On day 7–10 PBM, the females were dissected, and their guts were examined under a light microscope. The intensities and locations of infections were evaluated, as described previously [19].

### 2.5. Tissue Sampling and Fluorescence Detection of Leishmania Post-Mortem

Rodents were killed by cervical dislocation under anaesthesia. Both ears (inoculated and contralateral), both ear-draining lymph nodes, spleen, liver, paws, and tail, were removed and placed on Petri dishes. Fluorescence detection of *Leishmania* was done immediately post killing using the In Vivo Xtreme optical display (Bruker Biospin, Ettlingen, Germany) at the Center of Advanced Imaging Methods at the First Faculty of Medicine, Charles University. Excitation and emission wavelengths were 570 nm and 700 nm. *Leishmania* positivity was evaluated as the color difference against the negative controls, i.e., organs and tissues of uninfected animals. After that, half of all the left and right ears were used for parasite detection by flow cytometry. The second half of each ear and other organs were stored at −20 °C for quantitative PCR (q PCR).

### 2.6. Flow Cytometry and qPCR

The preparation procedure was identical for all rodent groups. Half of the ear was homogenized into small pieces in the 1 mL well on the plate. Then 500 µL saline, 42 µL liberase (Sigma-Aldrich, St. Louis, MO, USA), and 2 µL DNAse (Top-bio s.r.o., Vestec, Czech Republic) were added. The plate with the samples was incubated for 50 min (37 °C, 5% CO_2_). After stopping tissue digestion with 10% heat-inactivated fetal calf serum (FBS Gibco, Thermo Fisher Scientific, Waltham, MA, USA), the samples were filtered through fine gauze into a cold tube and centrifuged (8 min/175 G, 8 °C). The pellet was resuspended in 800 µL saline. 200 µL of the suspension from each sample was used for parasite detection by flow cytometry using the cytometer LRS II (BD Bioscience, Franklin Lakes, NJ, USA) and the software Flowlogic 7.3. Before measurement, Hoechst dye 33258 (Sigma-Aldrich, St. Louis, MO, USA) was added to each sample to identify dead cells. Before tissue analysis, the sensitivity of flow cytometry was tested and scaled with culture-derived *Leishmania* samples (with a range of 10^1^–10^8^ cells). The final results are expressed per the whole volume of the suspension and corresponding to half of the ear pinna.

For q PCR, extraction of total DNA from rodent tissues and sand flies was performed using a DNA tissue isolation kit (Roche Diagnostics, Indianapolis, IN, USA) according to the manufacturer’s instructions. Q PCR for detection and quantification of *Leishmania* parasites was performed in a Bio-Rad iCycler and iQ Real-Time PCR Systems using the SYBR Green detection method (iQ SYBR Green Supermix, Bio-Rad, Hercules, CA, USA) as described [20].

### 2.7. Statistical Analysis

Statistical analyses were carried out using the R software (http://cran.r-project.org/), and the results were graphically visualized using the “ggplot2” package in the same software. A *p*-value of < 0.05 was considered to indicate statistical significance. The relationship between response variables (weight, lesion size, number of parasites, percentage of infected organs, and infectivity to sand flies) and explanatory variable (infection with CDP or SDP) were tested for each rodent species separately. In the case of nonsignificant differences between the infection methods, the data were summarized for the subsequent analysis. The between-species differences were tested with the BALB/c set as the reference level for the variable rodent species. The continuous response variable (weight, lesion size) and their relationship with categorical (infection method, rodent species) and continuous (week p.i.) explanatory variables were examined fitting multilevel linear regression models (package “nlme”), taking into account the correlation between repeated measures of the same animal over time. The model used included the interaction term between categorical and continuous explanatory variables. The association between numbers of parasites in inoculated ear and infection method and between numbers of parasites in inoculated ear and rodent species (between-species differences) were analyzed with GLM with a negative binomial distribution. Wilcoxon signed-rank test was used to compare the numbers of parasites in inoculated ear measured by two methods, qPCR and flow cytometry. The test was run for each rodent species separately. The influence of the infection method (SDP, CDP- infection dose 10^5^ and 10^7^) on the percentage of *L. donovani* infected organs in rodent bodies was measured by 2- and 3-sample test for equality of proportions. The relationship between the percentage of sand flies infected by feeding on rodent host (infectiousness) and between percentages of sand flies infected by feeding on rodent host and rodent species (between-species differences) were tested with GLM with a quasi-binomial distribution.

### 2.8. Animal Experimentation Guidelines

Animals were maintained and handled in the animal facility of Charles University in Prague following institutional guidelines and Czech legislation (Act No. 246/1992 and 359/2012 coll. on Protection of Animals against Cruelty in present statutes at large), which complies with all relevant European Union and international guidelines for experimental animals. All the experiments were approved by the Committee on the Ethics of Laboratory Experiments of the Charles University in Prague and were performed under permission no. MSMT-38031/2016-2 of the Ministry of Education, Youth and Sports. Investigators are certified for experimentation with animals by the Ministry of Agriculture of the Czech Republic.

## 3. Results

### 3.1. Development of L. major in Four Rodent Species

In total, 12 BALB/c mice, 12 *P. sungorus*, 16 *C. griseus*, and 16 *L. lagurus* were infected; half of them with 1 × 10^5^ culture-derived parasites (CDP) selected with PNA for high representation of metacyclic forms and the second half with 6–7 × 10^4^ sand fly-derived parasites (SDP), where metacyclics comprised 69–73% of all morphological forms. The numbers of SDP were derived from 10 dissected sand fly females, without any adjustments, to keep the character of the inoculum as natural as possible.

All four rodent species showed stable growth of the body weight during the experiments, and no significant differences in weight gains were found between the groups infected with CDP and SDP (Appendix A). In addition, the lesion growth was very similar for the two inoculum types in BALB/c mice, *P. sungorus*, and *L. lagurus.* In *C. griseus*, lesions in the SDP inoculated group developed more slowly, but the difference against the CDP infected group did not reach the statistical significance (Figure 1 and Appendix A).

Lesions appeared very early in BALB/c mice (already on week 2 p.i.), one week later in *P. sungorus* and *L. lagurus* (in week 3 p.i.), and even later in *C. griseus* (week 4 p.i.). Compared to susceptible BALB/c mice, the growth of lesions was very similar in *P. sungorus* (*p* = 0.08); the lesions have an ulcerative character in both species (Figure 2a,b), and their size increased progressively until the end of the experiment. In some *P. sungorus*, the skin surrounding ears was also affected by redness and exuviation (Figure 2c). Experiments with BALB/c mice and *P. sungorus* were, therefore, finished on week 9–11 p.i. to avoid excessive distress to the animals. In *L. lagurus*, the lesion growth was significantly slower than in BALB/c mice (*p* < 0.0001), and although the character of lesions was initially also ulcerative (Figure 2d); their size increased only to week 7 p.i. and then began to reduce. These animals were able to resolve lesions by necrosis of affected parts, resulting in a reduction of the size of ear pinnae (Figure 2e). In *C. griseus*, lesion development was also significantly slower than in BALB/c mice (*p* = 0.0001); lesions were not ulcerative and were fully healed in some (3 out of 13) animals (Figure 2f, Appendix A).

Numbers of parasites in inoculated ears were evaluated from the samples taken post-mortem using two different methods—qPCR and flow cytometry. Similar to the previous analyses, qPCR did not reveal significant differences between animals inoculated with CDP and SDP; therefore, results were summarized for both infection modes. Parasite loads in inoculated ears generally ranged between 10^2^ and 10^7^ parasites with a median of 10^3^–10^4^ in *L. lagurus* and 10^5^–10^6^ in the remaining three species. Both quantification methods showed similar numbers of parasites in inoculated ears of BALB/c mice and *P. sungorus* (Table 1 and Appendix A). Based on the data from qPCR, significantly lower parasite loads in comparison with numbers in BALB/c mice were detected in *C. griseus* (*p* = 0.03), and the parasite load was lowest in *L. lagurus* (*p* = 0.0003). Based on data from flow cytometry, the difference between the four species was insignificant in this respect. The direct comparison of the two quantification methods in individual species revealed no significant differences in BALB/c mice (*p* = 0.6221), *P. sungorus* (*p* = 0.4357), and *C. griseus* (*p* = 0.08), while in *L. lagurus* qPCR gave significantly lower numbers than flow cytometry (*p* = 0.011).

The distribution of parasites in rodent bodies was evaluated using the PCR and the fluorescence detection with In Vivo Xtreme (Appendix A, Figure 3a, and Appendix B). In BALB/c mice, parasites remained restricted to both ears (inoculated and contralateral) and their draining lymph nodes. Only in one animal, the liver was also infected. In all remaining species, *Leishmania* parasites were detected in all the tested organs and tissues. In *P. sungorus*, the infection rates in all these tissues were the highest, and these hamsters were also the sole species where blood was also infected. Generally, the PCR was more sensitive than fluorescence detection. For example, numbers of parasites detected in contralateral ears by qPCR reached up to 86 thousand, 56 thousand, 31 thousand, and 15 thousand in BALB/c mice, *P. sungorus*, *C. griseus*, and *L. lagurus*, respectively, but the ears did not produce higher fluorescence signal than negative controls. In addition, the fluorescence detection was not applicable to densely haired paws of *P. sungorus* and *L. lagurus*, where even the control animals produced too strong a fluorescence signal (Appendix B). On the other hand, this method gave a good spatial picture of the parasite distribution. In ear pinnae, the fluorescence signal mostly corresponded to areas affected by skin lesions (Figure 3b). In the liver, the signal came either from the single site (typically in BALB/c mice or *L. lagurus*), or there were several smaller fluorescence centers dispersed over the whole organ (apparent in *P. sungorus* infected with CDP, Appendix B).

Infectiousness of animals, tested by xenodiagnosis experiments with *P. duboscqi* females, was, again, very similar in the groups infected with CDP and SDP (Table 2). Infectiousness corresponded very well to parasite loads in inoculated ears: The values in *P. sungorus* and *C. griseus* (29% and 11% of sand flies infected, respectively) did not differ significantly from infectiousness of BALB/c mice (14% of sand flies infected, *p* = 0.19 and *p* = 0. 54, respectively), while *L. lagurus* infected significantly fewer sand fly females than BALB/c mice (8% of sand flies infected, *p* = 0.006).

### 3.2. Development of L. donovani in Four Rodent Species

In total, 14 *C. griseus*, 12 BALB/c mice, 12 *L. lagurus*, and 12 *M. auratus* were infected with *L. donovani*; 3 *L. lagurus* and 3 *M. auratus* with 10^5^ CDP, three individuals of each species with 10^7^ CDP, and 32 animals with 5–63.9 × 10^4^ SDP, where metacyclics comprised 29–46% of all morphological forms.

All the rodent species showed stable weight gain during the experiment, and the weight did not differ between the CDP and SDP groups (Appendix A). None of the inoculated animals developed lesions or other external signs of the disease throughout the entire experiment. Nevertheless, qPCR performed at the end of the experiment (on week 30 p.i.) revealed the presence of *L. donovani* DNA in various tissues and organs of infected rodents (Appendix A). The size of the inoculum (10^5^ vs. 10^7^ parasites) considerably influenced the outcome of the infection. In both species inoculated with 10^5^ CDP (*M. auratus* and *L. lagurus*), only one of 3 individuals became infected, dissemination through the body was limited, and only low parasite loads (<10^3^) were found in tissues of these animals (Figure 4a). Higher infection rates and parasite loads were detected in animals infected with 10^7^ CDP and SDP, and no significant differences were found between these two groups (Appendix A). Therefore, the data from 10^7^ CDP and SDP inoculated animals were combined for further analyses.

In *M. auratus*, 7/9 animals infected with 10^7^ CDP and SDP maintained the parasite till the end of the experiment. However, *Leishmania* parasites were present in low numbers (<10^3^ per sample) and remained restricted to inoculated ears or, in one specimen, draining lymph nodes of the inoculated ear (Figure 4a). In BALB/c mice, 8/12 individuals showed infection and the parasite visceralized into various tissues and organs except for the tail and blood, but again, the parasite numbers did not exceed 10^3^. In *C. griseus*, *Leishmania* were detected in 12/14 animals and spread to all the tested tissues and organs except for the tail and blood. Contrary to *M. auratus* and BALB/c mice, parasite loads per sample were often higher than 10^3^ or 10^4^. All *L. lagurus* inoculated with 10^7^ CDP and SDP were *Leishmania*-positive, parasites were detected in all the tested tissues except blood, and high parasite loads (>10^4^) prevailed. Infected spleens showed distinct enlargement (Figure 4a,b).

Infectiousness to sand flies was tested from weeks 15 to 30 p.i. *Phlebotomus orientalis* females were allowed to feed on the whole body of anesthetized animals (Table 3). All the 514 and 207 *P. orientalis* females fed on BALB/c mice and *M. auratus*, respectively, were negative. On the contrary, for *C. griseus*, sand fly females became infected at week 25, and the infectiousness persisted until the end of the experiment in both 10^7^ CDP and SDP groups. In total, 144 *P. orientalis* females were used for xenodiagnosis with *L. lagurus*. In this case, considerable differences appeared between the three groups. Specimens inoculated with 10^5^ CDP were not infectious to the vector during the whole experiment, while in both remaining groups, sand flies became infected 25 weeks p.i. (25% of sand flies infected) and the infectiousness was still high (18.5%) or even increased (57.1%) 30 weeks p.i. in animals inoculated with SDP and 10^7^ CDP, respectively.

## 4. Discussion

Experimental infections of *C. griseus, L. lagurus*, and *P. sungorus* were performed to establish new laboratory models for studying leishmaniasis. Three Asian rodent species were compared here with the most widely used laboratory animals in leishmaniasis research—BALB/c mice and golden hamsters. All the tested Asian rodent species were highly susceptible to both *Leishmania* species, but showed different disease phenotypes. All Asian species showed high susceptibility to *L. major*, surpassing BALB/c mice with the wider anatomical distribution of the parasites and offering the model of both the healing phenotype (*L. lagurus* and *C. griseus*) and the non-healing phenotype (*P. sungorus*). *Lagurus lagurus* also exhibited excellent susceptibility to *L*. *donovani*, presenting giant parasite loads in all the tested tissues and organs.

Chinese hamsters, *C. griseus*, were already introduced to the study of leishmaniasis in 1924 by Smyly and Young [11]. All ten animals in their experiments were successfully infected with intraperitoneal inoculation of large amounts of liver and splenic emulsion from kala-azar patients. This study was continued, and in 1926, the authors described a high rate of *C. griseus* infections (129 from 182 animals), and although hepatosplenomegaly was developed, the animals remained in good condition up to a year [12]. Hindle and Patton showed, in the same year, that *C. griseus* were also susceptible to intraperitoneal inoculation of promastigote forms, while intradermal and subcutaneous inoculation of amastigotes or promastigotes were not successful [13]. In our experiments, initiated with intradermal inoculation of promastigotes, *C. griseus* was confirmed as highly susceptibility to *L. donovani—*86% of animals were successfully infected. Although no external signs of the disease were apparent, parasites disseminated to almost all tested organs and tissues with the highest presence in inoculated ears (79% of animals) and spleen (71% of animals). Infectiousness to sand flies was confirmed from week 25 p.i. In addition, *C. griseus* were also susceptible to *L. major*. The parasites persisted in inoculated ears in 88% of animals, disseminated to all tested skin samples (contralateral ears, paws, tail), and to a lesser extent, also reached visceral organs (13% and 31% of liver and spleen samples, respectively). The ear skin lesions were not ulcerative, well-tolerated, or even healed, and animals were highly infectious to sand flies, of the tested species most accurately mimicking the features of the natural host of *L. major*. In natural reservoirs, visceralisation of *L. major* was observed repeatedly—bone marrow, lymph node and spleen of *Psammomys obesus* and spleen and liver of *C. gundi* caught in Tunisia were identified as positive by ITS1-PCR [23,24] and spleen and liver of gerbils and *Rattus rattus* were shown to be PCR- positive in southern Iran [25,26].

Similar to *C. griseus, L. lagurus* also tended to heal skin lesions caused by *L. major*. Parasites were detected in 94% of inoculated ears and 44% of forepaws and were also present in other skin samples (contralateral ears, hindpaws, tail), but only exceptionally visceralized. Concerning VL, these rodents have been used as animal models in few studies performed in the 1990s in China (reviewed in [15]). Our results proved higher susceptibility of *L. lagurus* to *L*. *donovani* compared to both golden hamsters and BALB/c mice. High loads of parasites were detected in both external and visceral tissues and organs, and the infectiousness to sand flies was the highest among the tested species. Except for splenomegaly, the animals did not show signs of suffering, which favors *L. lagurus* as the optimal model for progressive disseminated visceral leishmaniasis.

Finally, *P. sungorus* showed exceptional susceptibility to *L. major.* The growth of skin lesions was progressive, similar to BALB/c mice and the parasite numbers in inoculated ears were also comparably high. Parasites disseminated to all tested tissues and organs, including blood and viscera (71% of spleen samples infected), and of all the tested species, the highest infection rates, and infectiousness to sand flies were observed. Therefore, *P. sungorus* may serve as a genetically polymorphic animal model for testing drug resistance or vaccine efficacy, an alternative to inbred lines of susceptible mice.

The three Asian rodent species share distribution with human pathogenic *L. donovani*, *L. infantum*, and *L*. *major*, but reports of their involvement in the life cycle of the parasites are lacking. To our best knowledge, the only indirect indication is that the related species *Cricetulus migratorius* was reported to be infected with *L. donovani* species complex in Iran [27,28]. In nature, the three rodent species included in this study prefer arid habitats like steppes, steppe forests, and semi-deserts. *Lagurus lagurus* are widely distributed from Ukraine to north-west China, *C. griseus* range from Kazakhstan and west Siberia to Mongolia, north-east China, and the Korean peninsula, and *P. sungorus* are restricted to Kazakhstan and south of west and central Siberia [29]. In theory, all species fulfill the criteria postulated by Ashford (1996) [30] for the reservoir host—in their habitats, they are abundant, forming a large proportion of the mammalian biomass during population outbreaks, and are sufficiently long-lived to allow *Leishmania* surviving through the non-transmission season. As shown in this laboratory study, their response to infection also meets Ashford’s criteria—a large proportion of individuals are infected; they remain infected for a long time without the acute disease, and the parasites are in the skin accessible to feeding sand fly vectors.

The development of *Leishmania* infection in the host body is a complex process that depends on many factors, the most important being the *Leishmania* species and virulence of the parasite strain on one side and secondly on the genetic background and the immune response of the mammalian host. In addition, the clinical manifestation of the disease is influenced substantially by the experimental design, especially the infection dose, type of inoculum (procyclic/metacyclic promastigotes or amastigotes), and inoculation route [8,31,32,33]. In the mouse model, it has been shown that the delivery of the parasites by sand flies makes a substantial contribution to *Leishmania* infection in comparison with syringe inoculation [34]. The significant influence of the inoculation route on disease outcome was also observed in the hamster model: Disease progressions were slower, when *L. braziliensis*, *L. mexicana, L. donovani* or *L. infantum* infections were initiated by a sand fly bite or intradermal inoculation. Infections initiated with intracardiac or subcutaneous inoculation showed unnatural rapid visceralisation without the formation of skin lesions [33,35,36].

Initiation of infection with sand fly bite comprises several factors exacerbating the disease progression, specifically, sand fly saliva components (reviewed in [37]), parasite-derived exosomes [38], and promastigote secretory gel [34]. However, using this natural infection route, the inoculum dose cannot be controlled—the numbers of parasites transmitted by sand fly bite differ in several orders of magnitude between individual females [39,40,41]. Therefore, we used natural parasite forms derived from thoracic midguts of experimentally infected sand flies with late-stage infection (SDP), which were homogenized, counted, and inoculated by syringe. The inoculum size of SDP (5–8 × 10^4^) in *L. major* did not exceed the upper numbers delivered to host tissues by *P. duboscqi* bite (1 × 10^5^) [39], and the percentage of metacyclic forms (69–73%) corresponded to the average percentage of metacyclic forms in thoracic midguts of *P. duboscqi* [42]. In *L. donovani*, the infectious dose of SDP was slightly higher (5 × 10^4^–6.4 × 10^5^), but in this case, a lower percentage of metacyclic forms was present in the inoculum (29–46%). The disease progression in animals inoculated with SDP was compared with animals infected with promastigotes from stationary-phase cultures (CDP). Homogenates of sand fly salivary glands were co-inoculated with both inoculum types to enhance the development of infection [43], and the intradermal route of infection was used to ensure parasite exposure to the local immune response in the skin [44]. No significant differences in lesion growth, infectiousness to sand flies, and parasite loads were found between animals infected with SDP or CDP in any rodent, and *Leishmania* species. These results suggest that culture-derived parasites with sufficient representation of metacyclic forms co-inoculated with sand fly saliva intradermally can adequately simulate the natural features of the disease.

On the other hand, the progression of *L. donovani* infections in *M. auratus* and *L. lagurus* was significantly affected by the infectious dose—lower infection rates, dissemination, and parasite loads were detected in rodents infected with 10^5^
*L. donovani* CDP compared to animals infected with 10^7^
*L. donovani* CDP and SDP. This is in accordance with previous studies with various *Leishmania* species using the mouse model [34,39,45,46].

Distributions and numbers of parasites in rodent bodies were evaluated from samples taken post-mortem using three different methods—quantitative PCR, flow cytometry, and the fluorescence detection with In Vivo Extreme. For this purpose, *L. major* and *L. donovani* parasites were marked with DsRed and RFP, respectively. Polymerase chain reaction (PCR) is being increasingly used for leishmaniasis diagnosis, characterization, and identification of *Leishmania* species, due to the high sensitivity for the detection and identification of *Leishmania* species (reviewed in [47]). Quantitative PCR is a commonly used method for determining the quantity of *Leishmania* DNA in various tissues of laboratory animals experimentally infected with *Leishmania* parasites, asymptomatic humans from endemic areas, and also for testing putative reservoir hosts [18,48,49,50]. Flow cytometry-based on the use of fluorescent probes is a relatively quick and cheap method that has been introduced to detect various protozoan parasites in the host cells [51,52,53,54]. When combined with the vital DNA Hoechst dye [55], this method is effective for visualizing both the living and dead populations of *Leishmania* parasites in the sample. In our study, both quantification methods were very specific and showed similar numbers of parasites in all rodent species, with the exception of *L. lagurus*. However, in this species, the preparation of samples for quantitative analyses was more difficult because of the much-reduced size of ear pinnae at the end of the experiment, which might influence the analyses.

In addition to PCR, the parasite distributions in rodent bodies were visualized by fluorescence detection with In Vivo Xtreme. In vivo imaging is a novel non-invasive procedure, which can be used for analyzing and monitoring small animal models of leishmaniasis [56,57]. If parasites were engineered to express luciferase, their numbers and distribution in individual animals could be imaged multiple times during longitudinal studies [56]. In this study, *Leishmania* parasites were marked with the red fluorescence protein suitable for flow cytometry, but the fluorescence signal with In Vivo Xtreme was not sensitive enough to detect parasites without killing of animals. Therefore, the analysis was performed post-mortem. While qPCR revealed parasite loads in infected organs, the fluorescence method provided valuable information about the *Leishmania* spatial distribution within these organs. Surprisingly, between-species differences were observed in this respect. In the liver, fluorescence signal came from the single spot in BALB/c mice and *L. lagurus*, while there were several smaller amastigote centers dispersed over the whole organs in *P. sungorus*. However, lower parasite loads were not detectable, and this method was not applicable to densely haired paws of *P. sungorus* and *L. lagurus*, due to high autofluorescence of the background.

Laboratory-bred golden hamsters are highly susceptible to visceral leishmaniasis [33,58]. They often exhibit severe clinical signs and symptoms during visceral infection (hypergammaglobulinemia, hepatosplenomegaly, anemia, cachexia) that is initiated by the intracardiac or the intraperitoneal route (reviewed in [59]). In this study, *L. donovani* infections were initiated intradermally, and parasites remained restricted to inoculated ears, in low loads. Mild disease outcome and a delay in disease progression in hamsters after intradermal inoculation of both *L. donovani* and *L. infantum* has been repeatedly described [60,61]. In addition, it has been demonstrated that the immune response to sand fly salivary protein protects hamsters against the fatal outcome of VL [62]. This study showed that, contrary to golden hamsters, the *L. lagurus* model enables the study of disseminated VL initiated via a natural intradermal route. Such a model is, therefore, closer to the active human disease and should be favored for drug and vaccine testing.

Inbred BALB/c mice are highly susceptible to *L. major* showing a non-healing phenotype associated with polarization to a Th2 response, which is not effective against intracellular parasites (reviewed in [63]). On the other hand, BALB/c mice are able to control *L. donovani* and *L. braziliensis* infection [64,65,66]. In this study, the tissue specificity was different for the two parasite species in BALB/c mice. *Leishmania major* amastigotes remained localized in inoculated ears and draining lymph nodes, without visceralisation. Mice infected with *L. donovani* showed spleen and liver infections in 58% and 33% of mice by week 30 p.i., respectively. This corresponds to general observation of murine visceral leishmaniasis with both *L. donovani* and *L. infantum—*the liver serves as the site for initial parasite expansion with fast parasite resolution, while in the spleen, visceralising *Leishmania* can persist for the life of the animal (reviewed in [31,59,67]). Although inbred mouse models have been invaluable for research on the principles of the mammalian immune response, wild rodents provide a more natural model of parasite-host interactions as, like humans, they are genetically polymorphic [6]. A larger spectrum of model species, covering the whole picture of human leishmaniasis, is also a prerequisite to facilitate vaccine development [68]. In this respect, *L. lagurus* and *C. griseus* may be used in research on the healing CL phenotype, while *P. sungorus* provides the non-healing CL phenotype. All three species showed higher dissemination of *L. major* in the tissues and organs than BALB/c mice.

This study provides experimental infections of three Asian rodent species to enrich the animal model spectrum available to both CL and VL research. *Cricetulus griseus, P. sungorus*, and *L. lagurus*, although all are susceptible to both *L. major* and *L. donovani*, differ in the infection outcomes, which enable their application to various research objectives. Further studies are necessary to better understand the immunological mechanisms involved in the response of these rodents to *Leishmania*.

## 5. Conclusions

1. Infections initiated with intradermal inoculation of 10^5^ culture-derived metacyclic *L. major* or with 10^7^culture-derived non-selected *L. donovani,* co-inoculated with 0.5 sand fly SG, were fully comparable in all disease manifestations to infections initiated with sand fly-derived parasites (10 sand fly thoracic midguts and 0.5 sand fly SG per animal).

2. *Lagurus lagurus* represented the healing phenotype with *L. major* infections. In addition, this rodent species showed higher susceptibility to *L*. *donovani* than golden hamsters and BALB/c mice—parasites were present in high loads in all the tested tissues and organs and the infectiousness to sand flies was the highest among tested species; therefore, these animals may best model disseminated visceral leishmaniasis.

3. *Cricetulus griseus* tolerated well or even healed *L*. *major* lesions, while parasites remained widely distributed through the body, and animals were highly infectious to sand flies, mimicking best the features of the natural host of the parasite. These rodents were also highly susceptible to *L. donovani*, similarly to *L. lagurus*.

4. *Phodopus sungorus* showed high susceptibility to *L. major* with the non-healing phenotype manifested in the progressive growth of ulcerative lesions and massive parasite loads in inoculated ears. Dissemination of parasites through the body and infectiousness to sand flies was even higher than in immunodeficient BALB/c mice.

5. Both qPCR and flow cytometry are highly sensitive methods for the evaluation of parasite loads in host tissues. Fluorescence detection using In Vivo Xtreme, although less sensitive and not suitable for accurate quantification of parasites, gives valuable spatial data for parasite distribution.

## Figures and Tables

**Figure 1 microorganisms-08-01440-f001:**
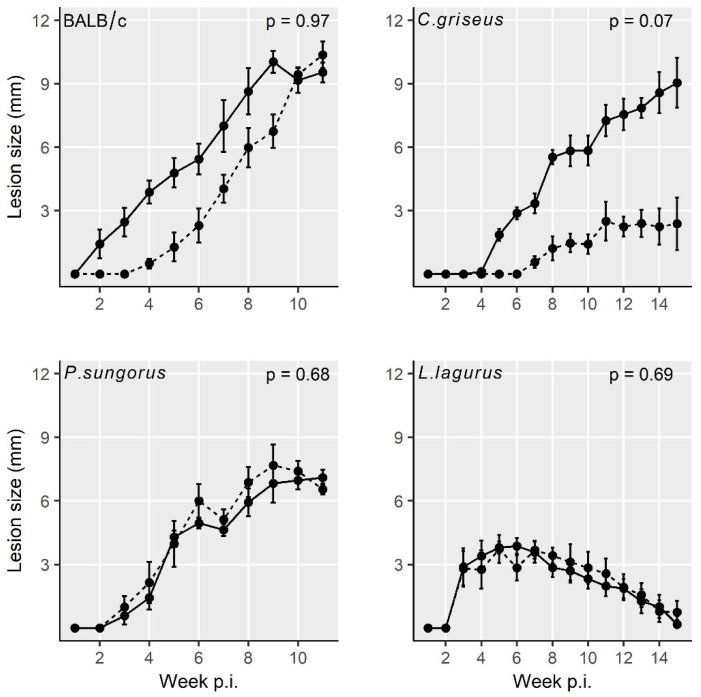
Lesion growth in animals inoculated with *L. major* using the two infection modes. Solid line = infection with culture-derived promastigotes (CDP); dashed line = infection with sand fly-derived promastigotes (SDP). *p* values indicate the statistical difference between the two infection types. Data are presented as the means ± standard errors of the means; 15% of the variance was explained by individual variability.

**Figure 2 microorganisms-08-01440-f002:**
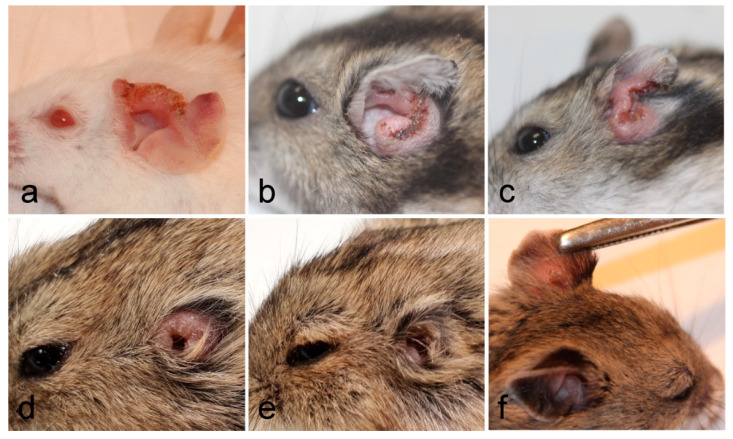
Inoculated ears of BALB/c mice (**a**), *P. sungorus* (**b**,**c**), *L. lagurus* (**d**,**e**), and *C. griseus* (**f**) showing external manifestation of *L. major* infections at the end of experiments.

**Figure 3 microorganisms-08-01440-f003:**
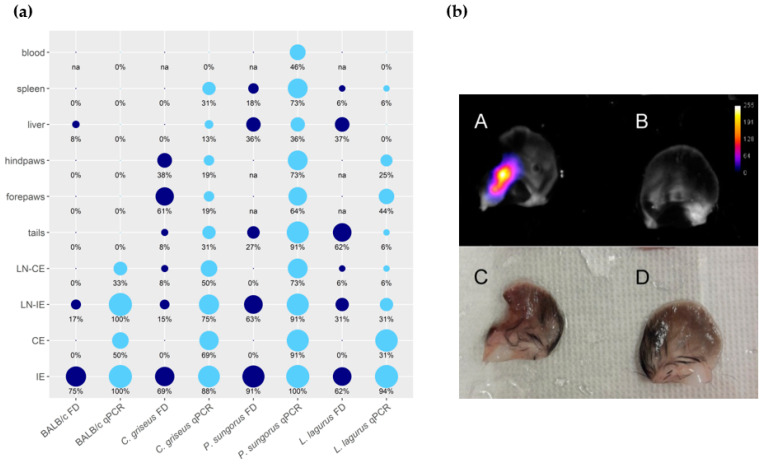
Anatomical distribution of *L. major* in four rodent species. (**a**) Results of fluorescence detection (FD, dark blue) and qPCR (light blue) presented in the balloon graph where the size of the balloon corresponds to the infection rate, i.e., the percentage of *L. major*-infected organs from the total sum of tested organs of the same type. IE = inoculated ears; CE = contralateral ears; LN-IE = draining lymph nodes of the inoculated ears; LN-CE = draining lymph nodes of the contralateral ears; na = not analyzed. (**b**) Ears of the *C. griseus* wetted with saline, photographed by the end of the experiment at week 15 p.i. A, B, images from In Vivo Xtreme optical display; C, D, images taken by Canon EOS 60D camera with Canon MP-E 65 mm f/2,8 1–5× Macro objective. A, C—inoculated left ear with ulcerative lesion and 2240 thousand parasites detected by q PCR. B, D—contralateral right ear with no external manifestation of the disease and 16.8 thousand parasites according to qPCR.

**Figure 4 microorganisms-08-01440-f004:**
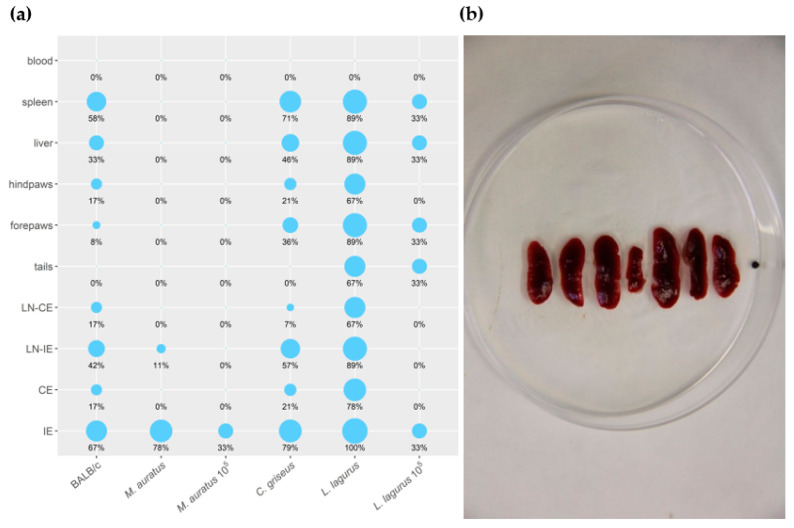
(**a**) Anatomical distribution of *L. donovani* in rodent species determined by qPCR. Results are presented by the balloon graph where the size of the balloon corresponds to the percentage of *L. donovani* infected organs from the total sum of tested organs of the same type. IE = inoculated ears; CE = contralateral ears; LN-IE = draining lymph nodes of the inoculated ears; LN-CE = draining lymph nodes of the contralateral ears. 10^5^: Animals inoculated with 10^5^ CDP. Animals that are not 10^5^–labeled were inoculated with 10^7^ CDP or SDP (**b**) Splenomegaly in *L. lagurus* infected with *L. donovani* in comparison with uninfected control animal (in the centre).

**Table 1 microorganisms-08-01440-t001:** Comparison of *L. major* numbers detected in inoculated ears by flow cytometry (FC) and qPCR (PCR). The numbers represent parasite loads in half of the ear pinna.

Rodent Species	No. of Samples		Median (in Thousands)		Minimum (in Thousands)		Maximum (in Thousands)		*p* Values ^1^	
	FC	PCR	FC	PCR	FC	PCR	FC	PCR	FC	PCR
BALB/c mice	12	12	430	438	2	0.3	2756	14,270	-	-
*P. sungorus*	5	11	732	293	28	46	820	2240	0.93	0.19
*C. griseus*	13	16	744	152	6	0	1947	2240	0.89	0.03
*L. lagurus*	16	16	7	3	0.04	0.1	6982	974	0.09	0.0003

^1^ significance of the differences against the values in BALB/c mice.

**Table 2 microorganisms-08-01440-t002:** Results of xenodiagnosis experiments performed with *P. duboscqi* and the four rodent species infected with *L. major*.

Rodent Species	Rodent Numbe ^1^	No. of Sand Fly Females	No. and (%) of Positive Females	Rodent Species	Rodent Number	No. of Sand Fly Females	No. and (%) of Positive Females
BALB/c mice *p* = 0.58	C1	30	0	*P. sungorus* *p* = 0.80	C1	33	5 (15)
	C2	32	7 (21)		C2	25	5 (20)
	C3	24	3 (12)		C3	22	6 (27)
	C4	30	4 (13)		C4	24	11 (46)
	C5	29	8 (27)		C5	22	8 (36)
	C6	30	5 (17)		**∑**	**126**	**35 (28)**
	**∑**	**175**	**27 (15)**		S1	32	6 (19)
	S1	32	6 (18)		S2	25	9 (36)
	S2	30	5 (17)		S3	33	9 (27)
	S3	31	8 (26)		S4	22	10 (45)
	S4	34	0		S5	23	6 (26)
	S5	39	5 (13)		**∑**	**135**	**40 (30)**
	S6	33	0		**Total**	**261**	**75 (29)**
	**∑**	**199**	**24 (12)**	*L. lagurus* *p* = 0.36	C1	4	0
	**Total**	**374**	**51 (14)**		C2	2	0
*C. griseus* *p* = 0.84	C1	23	6 (26)		C3	2	0
	C2	24	3 (13)		C4	20	2 (10)
	C3	27	2 (7)		C5	25	0
	C4	23	1 (4)		C6	23	1 (4)
	C5	23	0		C7	24	2 (8)
	C6	24	5 (21)		**∑**	**100**	**5 (5)**
	C7	20	0		S1	3	0
	**∑**	**164**	**17 (10)**		S2	2	2 (100)
	**S1**	28	0		S3	2	0
	S2	15	5 (33)		S4	22	2 (9)
	S3	26	0		S5	18	1 (6)
	S4	23	8 (35)		S6	13	3 (23)
	S5	15	2 (13)		S7	26	3 (12)
	S6	19	0		S8	22	0
	**∑**	**126**	**15 (12)**		**∑**	**108**	**11 (10)**
	**Total**	**290**	**32 (11)**		**Total**	**208**	**16 (8)**

^1^ CX, rodents inoculated with CDP, SX, rodents inoculated with SDP. *p*, the significance of the difference between CDP and SDP groups.

**Table 3 microorganisms-08-01440-t003:** Results of xenodiagnosis experiments performed with *P. orientalis* in the four rodent species infected with *L. donovani*.

Rodent Species	Week p.i.	No. of Animals Exposed			No. of Sand Fly Females			No. and % of Positive Females		
		10^5^ CDP	10^7^ CDP	SDP	10^5^ CDP	10^7^ CDP	SDP	10^5^ CDP	10^7^ CDP	SDP
BALB/c mice	10	-	4	4	-	21	32	-	0	0
	15	-	7	7	-	73	79	-	0	0
	20	-	4	4	-	55	76	-	0	0
	25	-	3	4	-	45	49	-	0	0
	30	-	2	4	-	25	59	-	0	0
	**∑**					219	295		0	0
*C. griseus*	10	-	4	4	-	11	7	-	0	0
	15	-	7	7	-	22	33	-	0	0
	20	-	4	4	-	11	28	-	0	0
	25	-	4	4	-	24	23	-	1 (4.2	1 (4.4)
	30	-	4	4	-	25	22	-	2 (8.0)	1 (4.6)
	**∑**					93	113		**3 (3.2)**	**2 (1.8)**
*M. auratus*	15	3	3	3	3	7	25	0	0	0
	20	3	3	3	26	20	NA ^1^	0	0	NA
	25	3	3	3	6	4	29	0	0	0
	30	3	3	6	13	14	60	0	0	0
	**∑**				48	45	114	0	0	0
*L. lagurus*	15	3	3	3	9	17	4	0	0	0
	20	3	3	3	14	3	NA	0	0	NA
	25	3	3	3	14	12	16	0	3 (25.0)	4 (25.0)
	30	3	3	6	14	14	27	0	8 (57.1)	5 (18.5)
	**∑**				51	46	47	0	**11 (23.9)**	**9 (19.1)**

^1^ not analyzed.

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
