# Peer review of "Central Asian Rodents as Model Animals for Leishmania major and Leishmania donovani Research"

_microorganisms, 2020, doi:10.3390/microorganisms8091440_

Round 1

Reviewer 1 Report

In this study by Vojtkova et al., the authors provided a detailed analysis of how Leishmania infection develops in three Central Asian rodents, with the goal of coming up with a model that would better recapitulate human disease. The authors tracked lesion development and parasite burden using a variety of complimentary techniques. They then comment on the particular advantages and disadvantages of the three rodents, compared to the more established models.

Although the results are interesting and appealing to the readership of this journal, the discussion needs to be a lot more focused on the primary objective of the manuscript.

Major concern:

  1. The discussion section of this paper needs to be restructured to better convey the importance of the work to the readers. As it stands, the discussion starts with a long essay on the epidemiology of leishmaniasis in Central Asian rodents. Instead, the discussion should start with a brief and poignant account of the main results and a statement of how the data supports the objectives of the study. In fact, the important part of the discussion (lines ~499-514) should appear much earlier, as that is what conveys the importance of the work. The authors should then discuss the results in the context of the literature. The rodent epidemiology section should be vastly summarized, and placed in the context of how the results compare to published data from the musculus / M. auratus models.

    I think that the authors should be clearer about the importance of this work to the field. After reading the Discussion/Conclusions, it was hard to decide why C. griseus, L. lagurus and P. sungorus might be better models than M. musculus and M. auratus in the context of L. major or L. donovani infection. Which one of these rodent models might better recapitulate L. major / L. donovani infection in humans and why?

Minor concerns:

  1. Image resolution (on-screen PDF and printed) appears quite low, and graph labels are hard to read because they appear somewhat fuzzy. To solve this, R has commands to export plots at 300 or 600 DPI resolution; the latter is preferred.
  2. On the figure that appears in line 569, ‘ly,ph’ should be replaced with ‘lymph’. The authors commented that they observed autofluorescence in non-infected animal paws due to the fur. There are gentle depilatory creams / gels that can be used to remove fur in the desired areas without harming the animal.

Author Response

We would like to thank Reviewer 1 for the positive evaluation of the manuscript and the helpful suggestions. We have aimed to meet all the requests.

 Major concern: 

  1. The discussion section of this paper needs to be restructured to better convey the importance of the work to the readers. As it stands, the discussion starts with a long essay on the epidemiology of leishmaniasis in Central Asian rodents. Instead, the discussion should start with a brief and poignant account of the main results and a statement of how the data supports the objectives of the study. In fact, the important part of the discussion (lines ~499-514) should appear much earlier, as that is what conveys the importance of the work. The authors should then discuss the results in the context of the literature. The rodent epidemiology section should be vastly summarized, and placed in the context of how the results compare to published data from the musculus / M. auratus models.

The discussion was restructured according to the recommendation. The brief account of the main results added in lines 387-392. The description of rodent distribution was substantially shortened and the ecology of the Asian rodents discussed in the context of their possible reservoir role (according to the request of the Reviewer 2), lines 438—444.

I think that the authors should be clearer about the importance of this work to the field. After reading the Discussion/Conclusions, it was hard to decide why C. griseus, L. lagurus and P. sungorus might be better models than M. musculus and M. auratus in the context of L. major or L. donovani infection. Which one of these rodent models might better recapitulate L. major / L. donovani infection in humans and why?

The discussion was completed in lines 521-523 and 533-540.

Minor concerns:

 Image resolution (on-screen PDF and printed) appears quite low, and graph labels are hard to read because they appear somewhat fuzzy. To solve this, R has commands to export plots at 300 or 600 DPI resolution; the latter is preferred.

The image resolution was raised to 600 DPI.

On the figure that appears in line 569, ‘ly,ph’ should be replaced with ‘lymph’.

The error was corrected

The authors commented that they observed autofluorescence in non-infected animal paws due to the fur. There are gentle depilatory creams / gels that can be used to remove fur in the desired areas without harming the animal.

Thank you for this suggestion; this may be helpful for follow up studies.

Reviewer 2 Report

The paper of Vojtkova and coll aimed at evaluating the susceptibility of three central Asian rodent species belonging of the Cricetidae family, towards L. major and L. donovani strains. External manifestation, lesion formation, parasite dissemination into various organs as well as the proliferation of Leishmania parasite were probed. In addition, the infectiousness towards sandflies was ascertained. From these experiments, the author deduces that these rodents might represent adequate animal models of L. donovani and L. major infection studies. In the study, a large part of the experimental procedure aimed at demonstrating the reservoir competence of these rodents is performed. These data are key elements to guide the quest of a wild reservoir of L. major and L. donovani parasite in endemic areas. Natural mammalian reservoirs for L. major belonged, for a majority of them, to the Muridae family (Rhombomys sp, Meriones sp, Psamomys sp, Gerbillus sp, Arvicanthis sp, Tatera sp, Nesokia sp), and to the Cricetidae (Microtus sp). Natural animal reservoir for L. donovani is less well known, homo sapiens being the primary reservoir. Data provided in this paper might pave the way of future research aimed at addressing the role of Cricetulus griseus, Lagurus lagurus, and  Phodopus sungorus in the epidemiology of visceral and cutaneous leishmaniases caused by L. donovani and L. major respectively.

Mesocricetus auratus is considered as a suitable animal model for L. donovani, mus musculus being considered as a model to address L. major infection. If the physiopathology and the immune response recorded in these animal models parallel those observed in humans, they, therefore, might provide some clues on their interest as animal models for drug and vaccine development.

General remarks

To introduce the interest of these animals as new models it would be important to discuss the limitation of the existing animal models, with reference on their interest for drug or vaccine development.  

What is the bioecology of Cricetulus griseus, Lagurus lagurus and Phodopus sungorus, and why they can be considered as potential reservoirs for L. major and/or L. donovani.

Technical remarks

  • I see that for sungurus and L. lagurus, lesion development appears identical in severity, whatever the initial inoculum or parasite source, or both!!. Why not using the same initial inoculum of SDP and CDP to perform a comparative experiment of these 2 sources of inoculum? Does variation in lesion development (rapidity and/or severity) might be related to the load of the inoculum?
  • Does the author have any information on the sensitivity of the flow cytometry methodology for parasite detection? Do results are expressed as a total number of events per volume (200ml) of suspension or per gr of tissue? If parasites express DsRed or GFP, why using Hoechst for parasite detection? If the author ascertained the number of viable parasites, it will be interesting to get an idea of the percentage of viable parasites in each tissue? Does it vary according to the host, or the parasite species?
  • I quoted lines 289 to 292 “Generally, the PCR was more sensitive than the fluorescence detection. For example, numbers of parasites detected in contralateral ears by qPCR reached up to 86 thousand, 56 thousand, 31 thousand, and 15 thousand in BALB/c mice, sungorus , C. griseus and L. lagurus , respectively, but the ears did not produce higher fluorescence signal than negative controls” Is nit possible that qPCR detect cell-free DNA circulation, that is a small amount of DNA that has escape degradation, as recorded in tumour ?

Author Response

We would like to thank Reviewer 2 for the report and the helpful suggestions. We have aimed to meet all the requests and provide the required information in the text of the manuscript.

General remarks

To introduce the interest of these animals as new models it would be important to discuss the limitation of the existing animal models, with reference on their interest for drug or vaccine development.  ¨

The discussion was enriched and the new text added to lines 521-523 and 533-540.

What is the bioecology of Cricetulus griseus, Lagurus lagurus and Phodopus sungorus, and why they can be considered as potential reservoirs for L. major and/or L. donovani.

The discussion was completed in lines 438-444.

Technical remarks

 I see that for sungurus and L. lagurus, lesion development appears identical in severity, whatever the initial inoculum or parasite source, or both!!. Why not using the same initial inoculum of SDP and CDP to perform a comparative experiment of these 2 sources of inoculum? Does variation in lesion development (rapidity and/or severity) might be related to the load of the inoculum?

The reviewer is correct that inoculum size is a very important factor influencing the disease outcome. In this study, we observed significant differences in infection rate, parasite load, dissemination and the infectiousness to sand flies between animals inoculated with 105 and 107 L. donovani (lines 334-338 in Results and 471-475 in Discussion). However, in experiments with L. major, the size of the inoculum was very similar - 6-7x104 in SDP (derived from 10 sand fly females) and 105 in CDP. Importantly, metacyclic forms prevailed in both inoculum types. Therefore, it is not so surprising that lesion development did not differ significantly between the groups infected with CDP/SDP. On the other hand, the between-species differences in lesion development must be caused by host-related factors as the different rodent species were inoculated in all experiment runs with the same SDP/CDP inoculum.

Does the author have any information on the sensitivity of the flow cytometry methodology for parasite detection?

Before tissue analysis, the sensitivity of flow cytometry was tested and scaled with culture-derived Leishmania samples (with a range of 101 – 108 cells). This information was added to Methods, lines 182-183.

Do results are expressed as a total number of events per volume (200ml) of suspension or per gr of tissue?

The final results are expressed per the whole volume of the suspension and correspond to half of the ear pinna (the sentence was added to Methods, lines 183-184) and the sentence “The numbers represent parasite loads in half of the ear pinna” was added to Results, line 285

If parasites express DsRed or GFP, why using Hoechst for parasite detection?

The Hoechst dye allows the easy and correct identification of dead cells (as written in Methods, lines 181-182); the aim was to exclude counting of damaged dead cells, still showing DsRed or GFP fluorescence.

If the author ascertained the number of viable parasites, it will be interesting to get an idea of the percentage of viable parasites in each tissue? Does it vary according to the host, or the parasite species?

The flow cytometry was applied only to infected ears. The representation of dead cells varied among animals of the same rodent species infected with the same parasite species more than between species, therefore, the between-species comparison was not provided.

I quoted lines 289 to 292 “Generally, the PCR was more sensitive than the fluorescence detection. For example, numbers of parasites detected in contralateral ears by qPCR reached up to 86thousand, 56 thousand, 31 thousand, and 15 thousand in BALB/c mice,sungorus ,  griseus and L. lagurus , respectively, but the ears did not produce higher fluorescence signal than negative controls” Is nit possible that qPCR detect cell-free DNA circulation, that is a small amount of DNA that has escape degradation, as recorded in tumour ?

PCR is a highly sensitive method and may reveal DNA from degraded cells. However, in these samples, the high amount of the DNA and the long time post inoculation (10-12 weeks) make this possibility unlikely. In addition, it has been published that Leishmania kinetoplast and nuclear DNA degradation occurs very rapidly after amastigote death (Prina et al. 2007, Microbes and Infection 9, 1307e1315). 

Reviewer 3 Report

General comments on manuscript No 925603:

The subject of this manuscript is within the field of interest for Microorganisms. This is a well-designed and conducted study in which the authors evaluate the susceptibility of three Asian rodent species Cricetulus griseus, Lagurus lagurus and Phodopus sungorus to two human pathogenic Leishmania species, Leishmania major and Leishmania donovani. The authors find that Lagurus lagurus is the optimal model for progressive disseminated visceral leishmaniasis caused by Leishmania donovani and that Phodopus sungorus shows a high susceptibility to Leishmania major. Also find that culture-derived parasites with enough representation of metacyclic forms coinoculated with sand fly saliva intradermally. The results presented in this excellent work are more robust than those obtained with methods used to date in this kind of studies.

This information is without a doubt relevant because enhance the animal model spectrum available to both cutaneous and visceral leishmaniasis.

The paper is well written and concise and is suitable for publication in Microorganisms, following some minor revision.

Specific comments:

  • Line 33: Replace “then” to “than”
  • Lines 82-83: move this sentence to the discussion.
  • Lines 94-95: Add a few words about how major and L. donovani strains were marked with DsRed and RFP red fluorescent markers.
  • Line 112: Defibrinated or anticoagulated rabbit blood?. Please, add information.
  • Lines 226-229: Why were used different methods of CDP and SDP metacyclics quantification?. Explain.
  • Line 477: Replace “Cricetulus griseus” to “C. griseus”
  • Line 499: Replce “L lagurus” to “ lagurus
  • Line 527: Replace “then” to “than”

Author Response

We would like to thank Reviewer 3 for the supportive evaluation of the manuscript and the helpful comments. We corrected and specifically addressed all the suggestions in the amended text of the manuscript.

Specific comments:

Line 33: Replace “then” to “than”

Corrected

Lines 82-83: move this sentence to the discussion.

Moved to the first paragraph of the Discussion

Lines 94-95: Add a few words about how major and donovani strains were marked with DsRed and RFP red fluorescent markers.

We added citation of the reference to the Methods (line 94) and the acknowledgement (lines 586-587)

Line 112: Defibrinated or anticoagulated rabbit blood?. Please, add information.

The information was added.

Lines 226-229: Why were used different methods of CDP and SDP metacyclics quantification?. Explain.

The numbers of SDP were derived from 10 dissected sand fly females, without any adjustments, to keep character of the inoculum as natural as possible.  The sentence was added to lines 232-234.

Line 477: Replace “Cricetulus griseus” to “C. griseus”

Corrected, now line 393

Line 499: Replce “L lagurus” to “lagurus”

Corrected, now line 415

Line 527: Replace “then” to “than”

Corrected, now line 554